# Towards a Didactic Concept for Heterogeneous Target Groups in Digital Learning Environments—First Course Implementation

**DOI:** 10.3390/jpm12050696

**Published:** 2022-04-27

**Authors:** Matthias Katzensteiner, Stefan Vogel, Jens Hüsers, Jendrik Richter, Oliver J. Bott

**Affiliations:** 1University of Applied Sciences and Arts Hanover, 30539 Hannover, Germany; oliver.bott@hs-hannover.de; 2Department for Medical Informatics, University Medical Center Göttingen, 37075 Göttingen, Germany; stefan.vogel@med.uni-goettingen.de (S.V.); jendrik.richter@med.uni-goettingen.de (J.R.); 3Health Informatics Research Group, University AS Osnabrück, 49076 Osnabrück, Germany; j.huesers@hs-osnabrueck.de

**Keywords:** didactic, Healthcare IT, citizens, E-Learning, digitalization, digitization, patient empowerment, education, communication, healthcare communications

## Abstract

In the context of the ongoing digitization of interdisciplinary subjects, the need for digital literacy is increasing in all areas of everyday life. Furthermore, communication between science and society is facing new challenges, not least since the COVID-19 pandemic. In order to deal with these challenges and to provide target-oriented online teaching, new educational concepts for the transfer of knowledge to society are necessary. In the transfer project “Zukunftslabor Gesundheit” (ZLG), a didactic concept for the creation of E-Learning classes was developed. A key factor for the didactic concept is addressing heterogeneous target groups to reach the broadest possible spectrum of participants. The concept has already been used for the creation of the first ZLG E-Learning courses. This article outlines the central elements of the developed didactic concept and addresses the creation of the ZLG courses. The courses created so far appeal to different target groups and convey diverse types of knowledge at different levels of difficulty.

## 1. Introduction

In times of the SARS-CoV-2 pandemic and, in perspective, climate change, online formats of classic face-to-face events are becoming increasingly relevant. In terms of access to information for all, methods such as E-Learning can be used for new ways of globalization [1]. With these methods, knowledge can be shared across national borders so that developing countries can also benefit [2]. Moreover, cultural exchanges may be an additional advantage through this approach, as well as sensitivity to differences in health care systems. While a face-to-face event allows direct human contact, online events can be offered and conducted independently of location and time. Of course, online events do not lack flaws and must therefore be used in a well-considered manner.

In the information age, data and information accumulate in great abundance and high frequency, so that non-specialist viewers quickly reach a capacity limit for information processing. In the media and in politics, complex facts are often strongly abbreviated or even distorted, as frequently observed during the COVID-19 pandemic in Germany, for example [3,4].

For non-specialist consumers of these media and political reports, it is no longer easy to process and retrieve the wealth of information. Due to the fast publication time on the Internet, many reports are disseminated at an early stage. In particular, study data of untested publications are already recited and used in a variety of media, which can lead to conflicts later on [5,6].

In order to be capable of classifying this high information density and distinguishing the quality of information, media competence and certain basic knowledge are required [7]. In order to enable the general population to understand certain complex facts on their own, it is necessary to build up abilities of adequate information handling and understanding. In particular, scientific definitions of terms, the basics of good scientific practice, and competence in the use of digital tools of any kind are of great importance here [8].

### 1.1. The Research Association

The *Lower Saxony Center for Digital Innovations* (in German: Zentrum für digitale Innovationen Niedersachsen, ZDIN) forms an interface between the Ministry of Economics and the Ministry of Science and Culture in Lower Saxony. The ZDIN is intended to promote networking and knowledge transfer between research and industry but, furthermore, make Lower Saxony’s research location visible to the general public. Within the framework of the ZDIN, several *future labs* have been developed that conduct research in different sectors. These are the future labs for *agriculture*, *mobility*, *healthcare*, *production and society*, *and work*.

The *Future Lab for Healthcare* (in German: Zukunftslabor Gesundheit, ZLG) of the ZDIN is, among others, involved in the planning and implementation of E-Learning programs. For this purpose, a special didactic concept for online-exclusive teaching was developed to serve as a guideline for educators to develop and implement pure online formats.

The ZLG particularly addresses the teaching of non-specialist audiences in Healthcare IT issues to achieve patient and people empowerment. Research and the latest findings in the ZLG’s research field shall be communicated to the general public. A focus should be especially on data privacy and data security, as skepticism regarding the trustworthiness of new technologies, i.e., of internet applications in general or AI technologies in particular, is widespread in society.

The ZLG is organized into three subprojects that address different aspects of Healthcare IT. Subproject 1 (TP1) is concerned with the development of a research data platform for shared data consolidation and research across site boundaries. The participating research sites of the ZLG are supposed to find the possibility to work together on research questions via this platform.

Subproject 2 (TP2) deals with the use of patient-related sensor technology for the improvement of everyday care. New sensor technology will be developed, and the use of established sensor technology will be optimized.

Subproject 3 (TP3), from which this publication arises, focuses on knowledge transfer of the research results based on E-Learning. During the project duration of the ZLG, a total of four online courses will be developed to transfer the research topics of subprojects 1 and 2. Further, TP3 is developing two demonstrators that address different aspects of online teaching.

### 1.2. Focus on Sub-Project 3 for Education, Training and Further Education

In TP3, the first demonstrator pursues AI-assisted physiotherapy training of shoulder patients via camera technology to monitor and evaluate training at home automatically. The combination of a special depth camera and smartphone application aims to provide gamification training, which allows patients regular but adequate health sport within home health care.

The second demonstrator develops a curriculum for statistical learning and data analytics in nursing professions and studies. This curriculum aims for integration into different study programs via ECTS allocation and thus is universally applicable. Teaching in demonstrator 2 takes place almost entirely online.

In addition to the content planned throughout the project, TP3 developed another course to introduce lecturers to online teaching. The so-called “Train the Trainer” course introduces participants to different tools and methods such as video production or evaluation concepts in addition to the actual didactic concept.

The ZLG-TP3 focuses on different target groups in order to test the broadest possible applicability of the developed concepts. The target groups addressed are patients and affected persons, citizens and the interested public, medical informatics and related professions, as well as representatives of the health care professions. In particular, the target groups of citizens and patients are considered a challenge since those target groups represent the cross-section of society and thus have a very high level of heterogeneity.

In this article, we present the courses designed and developed so far and compare their implementation. The already developed courses differ strongly from each other in structure and scope as well as their target groups but are based on the same didactic concept. The universal applicability of the basic didactic core concept for online teaching shall be shown and clarified by presenting the learning offers. Based on the experiences gained, we will continuously develop the concept and optimize its quality. For this purpose, evaluations are conducted by the participants at the end of the courses.

## 2. Materials and Methods

This publication relates to the paper “*Development of a Didactic Online Course Concept for Heterogeneous Audience Groups in the Context of Healthcare IT*” at pHealth 2021 [9] and explains further development stages. In the following, the didactic methods and approaches used for the course development are explained.

To achieve the goal of developing different online courses for various target groups, a concept was developed that covers relevant didactic aspects. These aspects are covered in different chapters:Target group definitionBasic concepts and scenarios of online didacticsTopics, learning objectives, and didactic scenarios of the ZLGNavigation concept to the online coursesDidactic core concepts for online course designDigital building blocks of learning management systems for course designConcept for course evaluation

To offer E-Learning, the ZLG uses and provides a learning management system (LMS) [10]. The ZLG compared and evaluated LMSs from different providers and thereafter decided to use ILIAS. The LMS serves as the central hub for the learning content, as the entire online teaching can be organized, structured, and accessed there. ILIAS stands for “Integrated Learning, Information and Work Cooperation System” in German and has been available since 2000 and was originally developed by the University of Cologne.

Overall, the basic functionalities of LMS are largely identical, so the differences lie particularly in the specific handling. Most LMSs, such as Moodle and ILIAS, support the SCORM (Sharable Content Object Reference Model) standard so that individual content can also be exchanged across LMS boundaries. The ZLG uses a shared ILIAS instance with the HiGHmed consortium teaching subproject to achieve a greater value from the collaboration [11].

The didactic concept also includes an evaluation concept that is used as a basis for iterative quality improvement of the developed E-Learning courses. Evaluation is intended to test the quality of educational programs, e.g., by surveying participants [12]. The evaluation concept of the ZLG teaching project distinguishes three dimensions: Type, method, and objects of evaluation.

The type of evaluation determines whether it is a formative or summative evaluation of the course. The evaluation method defines whether a quantitative or qualitative evaluation is to be conducted [13]. The evaluation object defines which unit is evaluated, such as course content, lecturers, or E-tivities.

From the aforementioned items, a core evaluation was designed to monitor the course quality of ZLG courses. Based on the standardized core evaluation, lecturers can extend the evaluation by individual aspects dynamically.

### 2.1. Target Group Definition

As already described in the introduction, the target groups predetermined in the project approach are characterized by great heterogeneity. In order to take this specificity into account, it must be possible to describe them as precisely as possible, using appropriate methods.

Following the system proposed in [14], the characteristics of the target groups were defined based on various criteria of socio-demographic, psychographic, and educational characteristics, as well as the expected educational behavior. The following attributes were described for each target group:Sociographic characteristics: Age structure, gender distribution, marital status, place of residence, area of influence, immigration background, level of education, employment status.Psychographic characteristics: Attitude, motivation, potential, strengths, weaknesses, aspirations, hopes.Educational characteristics/behavior: Previous education, media affinity, learning types, discussion types

Through the descriptions based on these characteristics, a precise differentiation of the target groups can be achieved, and the respective target group-specific communication and teaching can be designed.

For the target group of the interested public, a description based on these definitions is only possible in very general terms, as this group is characterized by the greatest possible diversity and heterogeneity. A clear description of the individual characteristics in a concise form is not feasible.

In order to at least approach an adequate description of the social structure, the ZLG didactic concept uses the marketing tool of the Sinus Milieus. This differentiates between varying subgroups, each of which has comparable attributes [15].

### 2.2. 5-Stage Model for Online Teaching by Gilly Salmon

The didactic core concept of the ZLG covers up to five phases and is essentially referring to the 5-stage model of Gilly Salmon for online teaching, which describes how online-exclusive teaching can be implemented with five successive steps (Figure 1). The model, according to Salmon, uses so-called “E-tivities”, which are designed to activate the audience and lead the participants to independent and autonomous learning [16].

The stages are designed to provide the audience group with new skills and confidence so that the challenge and intensity of learning can be increased with each stage. At the same time, communication between the audience is to be strengthened in order to promote group exchange and to discuss the increasing complexity of the tasks in the group. In stage 1, 1 to 1 communication between learners is initiated, as there are few other contacts at the beginning. In phases 2–4, the ratio changes to 1:n communication so that both the number of contacts and the frequency of exchanges increase. In the final phase, 5, the level of communication is reduced again in favor of individual follow-up so that each participating person can reflect on his or her own experience.

Stage 1, “Access and motivation”, is intended to enable participants to find a path around the learning environment, in this case, a learning management system (LMS). No learning context-related E-tivities and tasks are set, but low-threshold tasks are intended to motivate participation in the following work phases. Instructor support is important at this stage to resolve any access or operational issues with the LMS. Participants should perceive the course as a social environment and communicate transparently with each other.

Stage 2, “Online socialization”, allows for participants to interact with each other and explore the work environment. For this purpose, Salmon recommends that participants be considered under Wenger’s Communities of Practice concept [20], which includes three key components: *Shared undertaking*, such as distributed subtasks; *Reciprocity*, such as mutual acquaintance and trust; and *Shared repertoire*, which describes language, routines, or sensibilities. Socialization in the second stage is considered complete when participants communicate with each other without request.

Stage 3, “Information exchange”, initiates the joint processing and critical discussion of the learning material by the participants. The knowledge gained is to be applied in practical examples in order to work on a greater complexity of E-tivities in cooperation. The group dynamics should be increased at this level so that the interaction of learners is the focus. Learners should take participation in the course for granted and enjoy being active participants. Passive content transfer, such as reading texts, should be avoided, as this is considered detrimental to learning (lurking).

Stage 4, “Knowledge Construction”, should allow participants to build unconscious knowledge (tactical knowledge) to gain new perspectives. The E-tivities of this stage should demand critical, creative, and analytical thinking. Practical relevance and orientation are required in order to adequately apply the methods learned. The acquired knowledge patterns enable the learners to increase the learning effect in the long term. That way, what has been learned (theory) enters the subconscious and can be transferred to practical tasks (transfer performance).

Stage 5, “Development”, is designed to allow participants to make their own decisions about the learning content. A high level of autonomy is expected in this stage, as learners independently use the knowledge they have acquired to work on more advanced tasks. The audience intuitively collaborates at a high level without discussing group composition. By reflecting on the learning process, participants independently derive further knowledge and actions to use the knowledge in the future.

### 2.3. Didactic Core Concept for Online Learning in ZLG

A core didactic concept was developed at the ZLG to facilitate online teaching. The development and delivery of online courses are influenced by various factors. For the core concept, the focus was placed on the factors of group size, group dynamics and communication, learning and work phases, as well as the type and scope of supervision.

In relation to the factors identified, an adapted course structure was developed as a template (Figure 2), following the stage model of Gilly Salmon [16] and the HiGHmed teaching project [11,21] to support the development of online courses.

A special feature of the template is the usage of E-tivities. E-tivities, according to Salmon, are standardized in structure and consist of an introduction, objective, task, and discussion. The E-tivities are intended to pursue a constructivist approach to learning, ranging from the pure exchange of information to knowledge construction and personal development.

For developing courses in ZLG, a 5-Phase-Model was generated. This model is characterized by five learn-phases, which can be highly associated with Salmons 5-Stage-model. It should be emphasized here, however, that Salmon’s stages are not just reflections of the ZLG phase model. In particular, Salmon’s 5-stage model defines the activation of learners in the online environment at a different level, while the ZLG phase model defines concrete phases of work.

Phases 1–3 of the ZLG concept can be associated with the overall 5-step model according to Salmon, as the prerequisites for successful interactive knowledge transfer are created there.

Phase 1, “Welcome and introduction”, briefly introduces the course topic and the teachers in order to offer the participants a motivating introduction to the course. The expectations and level of knowledge of the participants are queried to establish an overall impression of the group. The course content should be appropriate to the level of the audience and can be supplemented with sources and references for missing knowledge. Finally, the technical and organizational issues of the participants are discussed to ensure full accessibility and usability of the Learning Management System.

Phase 2, “Scheduling and learning objectives”, introduces learning contents as well as learning objectives and the general course procedure. In this phase, possible later examination achievements should also be explained, and the communication options presented. Netiquette should also be provided and explained analogously to Salmon’s model.

Phase 3, “Elaboration of ***n*** learning units”, represents the actual knowledge transfer in the ZLG learning phase model. In this phase, the learners are taught the learning content and competencies over several learning units, whereby this phase is characterized by discursive reflection and practical, ideally collaborative application. The learning units of this phase should be similar in sequence and structure:Giving and activating input,Work on one or more tasks/questions,Formative assessment and feedback.

As the titles of the subdivision show, basic information introduces each learning unit on the respective focus (1). This is followed by one or more tasks and/or questions on the respective focus in order to internalize the theoretical knowledge (2). At the end of the learning unit, a formative assessment is carried out so that the learners can assess themselves, and the teachers can use the results to see whether the participants have achieved the learning objective (3).

Phase 4, “Summative assessment”, plays a significant role and dictates if a certificate is to be issued for successful participation in the course. Learning objective assessments serve as proof of whether or not and to what extent participants have achieved the learning objectives and acquired or deepened new competencies. While formative assessments are primarily used for the interim assessment of learning success, summative assessments are used in particular for the final assessment of participants’ performance.

Phase 5, “Conclusion and evaluation”, should offer participants the opportunity to evaluate the course. The instructors can derive quality assurance or quality enhancement measures from these evaluations. The evaluations of the participants can be used to generate comparisons between the learning success and the mood of the participants. Through the combination of quantifiable learning success (assessments) and qualitative statements (evaluation), insights into the quality of the learning offer can be gained. The ZLG has developed its own evaluation concept for carrying out evaluations.

## 3. Results

The individual subproject groups of the ZLG had the task of using the didactic core concept to create learning courses that address the subprojects’ research fields. The courses were created in the shared instance of the LMS ILIAS, as described above.

The following tables provide a brief overview of the created courses, regarding the criteria: Title, Content, Target Group, Learning Goals, Duration, Workload, Workload in synchronous Phase, Workload in asynchronous Phase, Amount of E-tivities, Learning Videos, other special features of the course.

### 3.1. ZLG-Metacourse “Train the Trainer”

The ZLG Metacourse, “Train the Trainer”, for knowledge transfer in ZLG, is structured by learning weeks. Each week aims for specific learning goals with intersecting topics. Following the ZLG didactic core concept, the level of difficulty increases steadily per week. The ratio between information transfer and self-learning is shifting continuously in favor of self-learning aspects (Figure 3).

The goal in the first week of the ZLG Metacourse is to enable the participants to use the learning platform for knowledge acquisition and exchange, but also to connect on a social level with other participants and the lecturers. For this reason, the E-tivities are aimed at demonstrating the features of the learning management system while introducing the individual virtually to the learning group. The introduction of the participants in the first week initially addresses the first three stages according to Gilly Salmon and the ZLG didactic core concept. The following weeks build on this preliminary work and then focus on phases three to five, not only to absorb information but also to apply it and further develop knowledge as a group.

In week two, the core didactic concept of the ZLG is taught. The aim of the E-tivities is to acquire knowledge individually with the support of learning videos and literature and then to create own E-tivities along with the phase model. In a final step, the E-tivities are refined by the group of participants and improved with the help of digital building blocks with regard to the motivation of the target group.

Week 3 focuses on the administrative structure of an online course, addressing the conceptual aspects and functional possibilities, in particular, the features that support the motivation of the participants. Another aspect of week 3 is the development of learning videos to provide target group-specific content according to the didactic concept.

In the last week of the course, the core evaluation concept is taught, which can be used in the ZLG and extended for individual courses. In this way, a level of comparability between the ZLG courses is created without ignoring the individuality of the courses. The ZLG “Train the Trainer”-Metacourse is concluded by the completion of the evaluation.

### 3.2. ZLG-Course “The Learning Healthcare System: How It Learns”

The course, “Learning Healthcare System: How it Learns—Secondary Data Use of Clinical Data for Medical Research”, is particularly structured by topics. Each topic deals with a self-contained subject area. The course structure and the course sequence are not related to temporal relationship but are determined by the topics that build on one another. The complexity of this course will not be successively increased, as it addresses society as a whole in particular. The course focuses primarily on explaining the basics and clarifying simple connections in the research area (Figure 4).

The course is supplemented by two synchronous meetings. There is one synchronous online meeting for the introduction to the course and one synchronous meeting for wrapping up the learned content. The majority of the learning content is provided asynchronously.

The first synchronous online meeting gives a brief introduction to the course and explains what the audience is about to discover in the following weeks. A case study of a patient is outlined for the participants to learn the context of the knowledge imparted.

The following lesson covers the basics concerning data itself and teaches the differentiation between data, information, and knowledge. Additionally, different kinds of data are described, such as structured and unstructured data. Different E-tivities provide the possibility to discover different datasets to learn differences and gain practical insights.

The next lesson introduces the challenge of interoperability and proposes different solutions such as terminologies or classifications. To outline the different possibilities, participants learn basics about the benefits of using standards such as ICD, OPS, or SNOMED CT.

In order to establish an overall view, the last lesson summarizes the knowledge gained beforehand in a synchronous online meeting.

### 3.3. ZLG-Course “Patient-Oriented Sensor Systems in Nursing: Application and Outlook”

The ZLG-Course Patient-oriented Sensor Systems in Nursing: Application and Outlook focuses on introducing participants to the principles of sensor technology, the possible application of this technology in healthcare, and the fundamentals of data processing and analytics.

High school students were selected as the target audience, thus forming a subset of the interested public. Participants have the choice of taking the course as individuals or as part of a group/class involving their teacher.

The course has therefore been designed to run either in parallel with school lessons, with a duration of five weeks, or as a block course with a duration of one week. The choice of different structuring can benefit participants with lower intrinsic motivation and less experience in self-guided learning (Figure 5).

The course content covers the basics of sensing, vital sign measurement, and the underlying IT technology. This knowledge provides a background for learning selected use cases of sensing in nursing. For each use case, a reference is provided to everyday sensor interaction.

Different types of sensors are covered. Among others, sensors for motion detection such as pedometers and vital sign measurements with electrocardiography or pulse oximetry are presented. In addition, sensors for the detection of environmental parameters, such as temperature, humidity, and fine dust measurement, are also discussed.

In addition, interactive elements, such as small tests and interactive videos, are offered alongside E-tivites and learning modules to ensure the achievement of the intended learning objectives. Optional E-tivities with advanced practical tasks allow highly motivated participants to gain hands-on experience in data analysis.

### 3.4. ZLG-Demonstrator 2: “Learning Health System in Action: Clinical Data Analytics”

The course Learning Health System in Action: Clinical Data Analytics differs from the courses for the public in that it is a classic curricular approach. The course aims to support health professionals across disciplines, such as nursing, physiotherapy, and alike, in applying and understanding analytic data procedures used for clinical data. This topic is embedded in the broader paradigm of the Learning Health System, where clinical data analytics and clinical decision support play a major role. The participants receive three ECTs when they have successfully completed the course (Figure 6).

The course is designed as a hybrid course, combining online educational resources and E-tivities with a synchronous session. As such, the course starts with a two-day kick-off session, followed by an online phase. The course ends with a final workshop.

The initial kick-off session is designed to introduce the course content, the students, and the learning platform. Furthermore, lectures are given to introduce the topic, i.e., the learning health system and the role of clinical data analytics therein. This kick-off session is then followed by a seven-week online course, in which each session is built on the previous in terms of content. The complexity of the content taught increases with each learning unit. The E-tivities during the online phase comprise tasks such as video lectures, completing quizzes, applying the acquired knowledge such as data analytic exercises, and peer-reviewing the co-students’ work. The course closes with a synchronous workshop, in which the focus lies on applying required data analytic skills in teamwork. This task uses realistic scenarios and clinical problems. For example, students are asked to develop a simple clinical decision system and present this system to their peers. The course closes with the course evaluation based on the concept of the ZLG, which itself is part of improving the course content and organization.

## 4. Discussion

The development of the first courses has been completed. During the set-up phase, a variety of other medical informatics fields were identified that could be taught. Based on the course evaluations, we want to evolve the developed courses longitudinally and iteratively (cf. [12,13]). Furthermore, the feedback supports the future workplan in regards to creating new courses and content, especially to address the interested public, and thus advance the transfer of knowledge from research to society. For the first rollout steps of the courses, we could learn that an essential pillar of success is the widespread advertising of the course offerings.

For very heterogeneous target groups that represent the entire spectrum of society, an efficient advertising strategy is of great value. For this reason, it is important to use a wide variety of communication channels in the future. For example, it would make sense to link up with schools in order to generate enthusiasm for exciting topics of medical informatics at an early stage and thus promote young talents. Furthermore, we see great opportunities in opening up the didactic concept and thus connecting further ZDIN labs to our course offering in order to provide a broader portfolio of research topics for potential participants.

The didactic concept developed could already be used as a guideline for the development of the four courses presented. As presented in the results chapter, various courses could be developed using the concept. Although the resulting courses differ greatly from each other both in terms of the target groups addressed, and the depth of content of the topics covered, they are based on a recognizable substantiated didactic concept.

While the “Train the Trainer” course trains lecturers, in particular, the TP1 and TP2 courses address secondary school students and the interested public. However, as the development of demonstrator 2 shows, the didactic concept can also be used to create a curricular learning offer for students.

These course developments demonstrate the adaptability and scalability of the didactic concept. The methodology is also sufficiently general that it can also be used in a research area independent of our own. The development of courses in research fields outside of health care or medical informatics should be examined to prove this thesis.

The first run of the courses could already be carried out with some participants. The quantitative impact of the evaluation results is too low at the moment, but further adaptations of the courses will follow in order to expand and optimize the course content. However, the experiences had with the implementation of the evaluation have already shown that it can be useful to position surveys very prominently and, if necessary, make them a prerequisite for certification. We expect that the number of participants will continue to scale as the courses become more established and advertising increases.

## 5. Conclusions

The developed courses address complex and eHealth-specific topics such as interoperability or data management and analysis in the healthcare sector, and the evaluation of the courses is intended to reveal the extent to which the right didactic concepts have been found. We plan to iteratively evolve the existing E-Learning courses and build additional courses incorporating eHealth research in the Future Lab for Healthcare, as it has already been started with the first courses.

At the time of publication, the didactic concept is available in version 1.0. In the project plan of the ZLG, an experience-based revision of the concept is intended so that after the implementation of the courses for TP1 and TP2, the evaluations of the participants and the experiences of the lecturers will be included in the concept.

At this point in development, the current approach appears to be well suited to fulfill the requirements for the development of E-learning courses for heterogeneous target groups.

Our approach might be helpful for other online education projects. Course design outside our own professional discipline could assess the applicability of our concepts.

As the immediate idea, the development of learning programs in the expanded ZDIN could be tested by the other future labs. This would allow to test and evaluate the scalability to other disciplines and their target audiences. The learning platform for course development is already being used by both the ZLG and the HiGHmed teaching project; thus, the integration of further future labs would be easily possible.

As explained in the introduction, knowledge transfer, in general, is facing major challenges. E-Learning opportunities are not only becoming more relevant due to crises such as the COVID-19 pandemic or climate change, but also the relevance of digital competencies of the population is increasing. Data literacy and patient empowerment are important milestones on the road to a digitally educated society.

In this context, the fifth statement of the Expert Council of the German Federal Government on COVID-19 of 30.01.2022 [22] (See Appendix A for a paraphrased translation or if no longer available online.) underlines the relevance of this topic. The experts emphasize that the loss of confidence in political decisions in the context of the management of the COVID-19 pandemic has increased sharply in recent months. They point to several problems and describe various aspects that increase the uncertainty of the population and thus provide room for misinformation and disinformation. Among other aspects, the experts highlight that the decision-relevant information should be translated into target group-specific language. With a focus on the informational fairness of different educational prerequisites, cultural and linguistic backgrounds, and age-dependent differences, it should be possible to adapt the information that is conveyed individually. For this purpose, the experts demand that the communicative content should be distributed via adequate channels, such as social media, e-health, or m-health offers. In addition, it makes sense to train different multipliers so that experts can offer competent knowledge transfer in direct contact with those affected. In summary, the experts state that sustainable communication structures should be established. An infrastructure for risk and health communication is a possibility to bundle existing competencies and push the translation of professional knowledge into a language that is appropriate for the target group.

The didactic concept introduced in this contribution demonstrates the feasibility of E-learning offerings for the general public. Through the use of LMS, insights and knowledge can be passed on to society. In this context, the developed concept makes it possible to use different target group-specific methods of communication across all disciplines. We are positive that the methods presented met the Expert Council’s suggestions, at least to some extent [22]. Further work progress has a high potential to meet the above-described challenges.

Complementary to this, E-Learning offerings may be factors for health and safety, as in the COVID-19 pandemic, and for sustainability, through emission avoidance.

Experienced teachers repeatedly emphasize that the E-Learning practiced today cannot completely replace traditional face-to-face teaching. Since we communicate by more means than just language, the further development of “classic” E-Learning approaches in the sense of virtualized reality is more than desirable in order to enable communication and interaction with all senses.

## Figures and Tables

**Figure 1 jpm-12-00696-f001:**
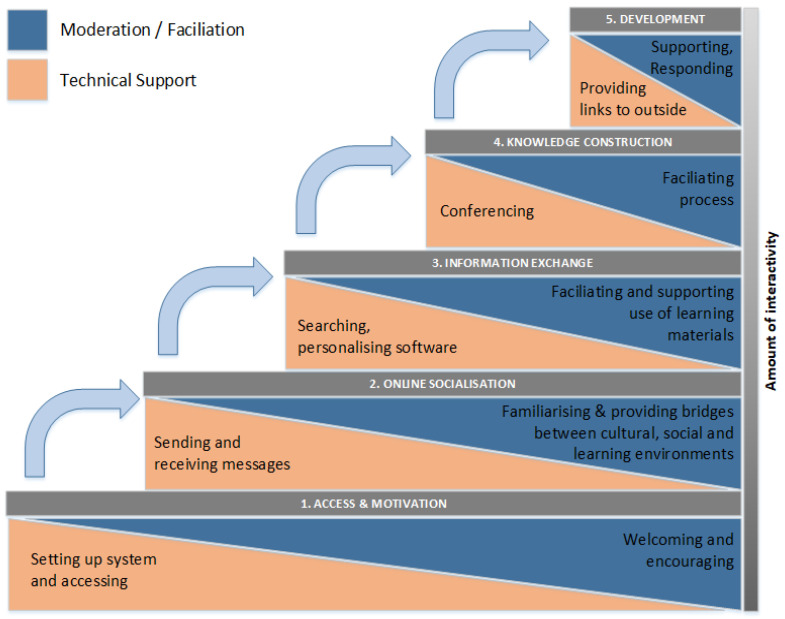
Salmon’s five-stage model for online learning. Gilly Salmon–E-tivities: The key to active online learning (2013) according to [16,17,18,19]. This image is available under the Creative Commons Attribution-NonCommercial-NoDerivatives 4.0 International (CC BY-NC-ND 4.0) license as stated in the publication and on the website of the author: https://www.gillysalmon.com/contact.html (accessed on 9 March 2022).

**Figure 2 jpm-12-00696-f002:**
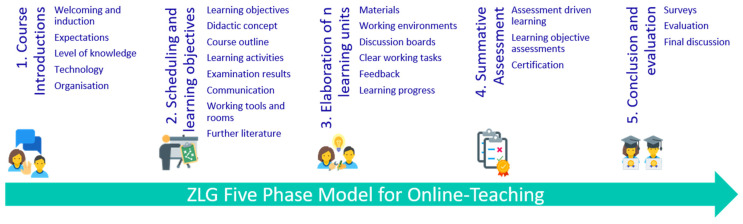
ZLG Five Phase Model for Online-Teaching based on [11,16,21].

**Figure 3 jpm-12-00696-f003:**
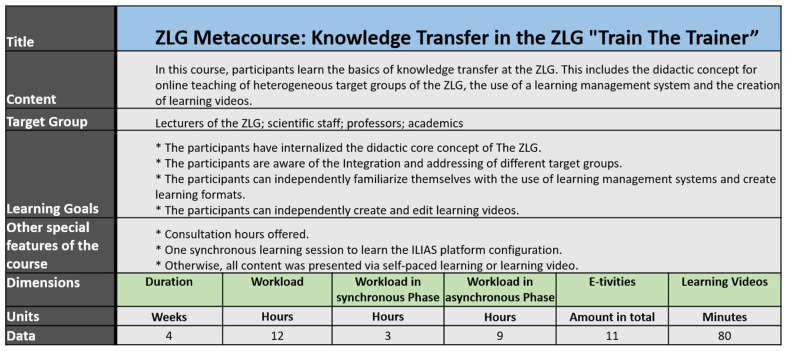
Course properties for ZLG-TP3 Metacourse: “Knowledge Transfer in the TLG “Train the Trainer”.

**Figure 4 jpm-12-00696-f004:**
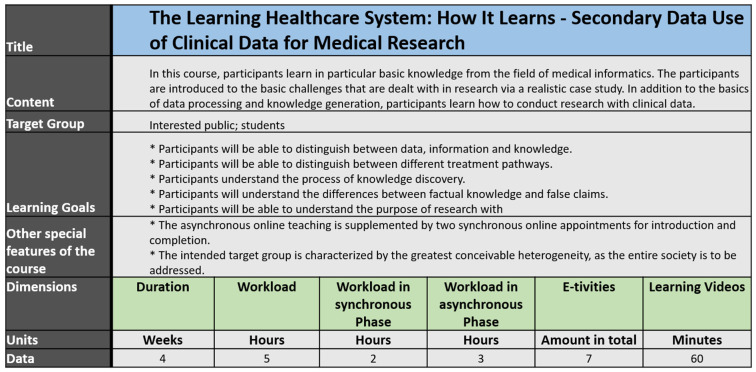
Course properties for ZLG-TP1 Course: “The Learning Healthcare System: How It Learns—Secondary Data Use of Clinical Data for Medical Research”.

**Figure 5 jpm-12-00696-f005:**
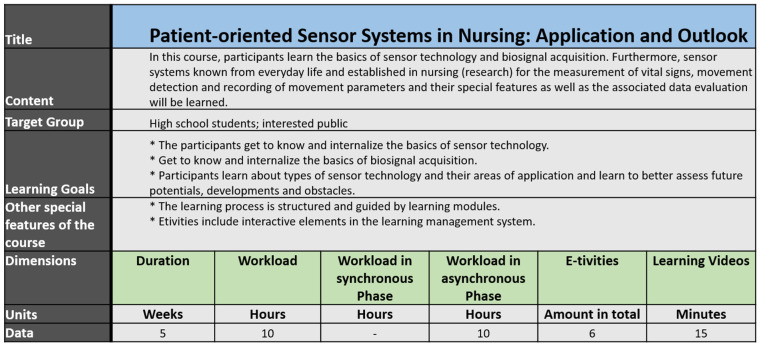
Course properties for ZLG-TP2 Course: “Patient-oriented Sensor Systems in Nursing: Application and Outlook”.

**Figure 6 jpm-12-00696-f006:**
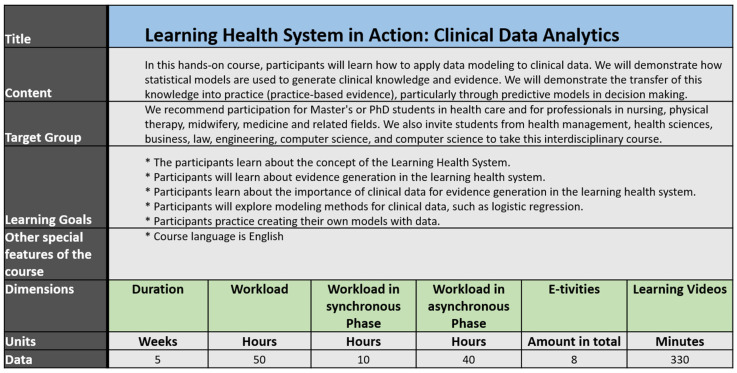
Course properties for ZLG-TP3-Demonstrator 2: “Learning Health System in Action: Clinical Data Analytics”.

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
