# Peer review of "Towards a Didactic Concept for Heterogeneous Target Groups in Digital Learning Environments—First Course Implementation"

_jpm, 2022, doi:10.3390/jpm12050696_

Round 1

Reviewer 1 Report

The authors explain well the concept of the E-learning method and phase model. 

Interestingly, the concept is novel and attractive for educators in the COVID-19 period focused on E-learning, not face-to-face education.

I recommend it as acceptance in this version.

Thanks a lot.

Author Response

Thank you very much for reviewing this submission. We appreciate your favorable review.

We agree with your sentiment that e-learning has been of greater importance since the COVID-19 pandemic. E-learning can continue to be an important factor in the future, including sustainability.

Thank you very much.

Reviewer 2 Report

I suggest adding comments relating to the use of ILIAS.
I suggest a few minor English language changes:
row 18 appeal different     appeal to different
row 31 independent of         independently of
row 169 referring the             referring to the

Author Response

Thank you for reviewing this submission. We appreciate your helpful feedback and are delighted with the positive assessment.

Regarding ILIAS, we have added further information and we have also made the language adjustments you recommended.

Thank you very much! 

Reviewer 3 Report

The paper is well structured and well written. It basically describes a methodology to develop e-Learning courses and provides details of some developed courses.

However, it is not clear what is the added value in the area of eHealth, since even though some of the example courses are on eHealth related topics, the scientific added value is on e-Learning.

One way of solving this limitation could be to identify the specificities of developing online courses for eHealth related content.

A minor detail on the content: Section 2.2 seems too detailed for describing an external model that is however used in further sections.

Author Response

Thank you for your feedback on our submission.

In order to emphasize the relation to eHealth, it is now pointed out that the developed courses will be further evolved iteratively. Furthermore additional courses will be built up, which will address further eHealth research contents of the Lower Saxony Health Future Lab as it has already been started with the first courses.

Thank you very much!

This manuscript is a resubmission of an earlier submission. The following is a list of the peer review reports and author responses from that submission.